# Epidemic efficacy of Covid-19 vaccination against Omicron: An innovative approach using enhanced residual recurrent neural network

Rakesh Kumar [1], Meenu Gupta [1], Aman Agarwal[1], Anustup Mukherjee[1], Sardar M. N. Islam[2]*

1 Department of Computer Science and Engineering, Chandigarh University, Punjab, India, 2 ISILC, Victoria University, Footscray, Australia

* sardar.islam@vu.edu.au

**Data Availability Statement:** The link to the dataset is: https://data.humdata.org/dataset/novel-coronavirus-2019-ncov-cases.

## Abstract

The outbreak of COVID-19 has engulfed the entire world since the end of 2019, causing tremendous loss of lives. It has also taken a toll on the healthcare sector due to the inability to accurately predict the spread of disease as the arrangements for the essential supply of medical items largely depend on prior predictions. The objective of the study is to train a reliable model for predicting the spread of Coronavirus. The prediction capabilities of various powerful models such as the Autoregression Model (AR), Global Autoregression (GAR), Stacked-LSTM (Long Short-Term Memory), ARIMA (Autoregressive Integrated Moving Average), Facebook Prophet (FBProphet), and Residual Recurrent Neural Network (Res-RNN) were taken into consideration for predicting COVID-19 using the historical data of daily confirmed cases along with Twitter data. The COVID-19 prediction results attained from these models were not up to the mark. To enhance the prediction results, a novel model is proposed that utilizes the power of Res-RNN with some modifications. Gated Recurrent Unit (GRU) and LSTM units are also introduced in the model to handle the long-term dependencies. Neural Networks being data-hungry, a merged layer was added before the linear layer to combine tweet volume as additional features to reach data augmentation. The residual links are used to handle the overfitting problem. The proposed model RNN Convolutional Residual Network (RNNCON-Res) showcases dominating capability in country-level prediction 20 days ahead with respect to existing State-Of-The-Art (SOTA) methods. Sufficient experimentation was performed to analyze the prediction capability of different models. It was found that the proposed model RNNCON-Res has achieved 91% accuracy, which is better than all other existing models.

## 1. Introduction

The first case of COVID-19 was encountered in Wuhan city, China, in Nov 2019. This deadly disease had a very bad impact on human life, and its fallout affected the socio-economic

**Funding:** The author(s) received no funding for this study.

**Competing interests:** The authors declare that they have no conflicts of interest to report regarding the present study.

conditions around the world. Before being declared a pandemic by the World Health Organisation (WHO), the Public Health Emergency of International Concern (PHEIC) was announced on 11[th] March 2020 [1]. The pandemic was caused due to respiratory viruses such as SARS-CoV-2 [2]. In India, the first case of COVID-19 was reported in Thrissur, Kerala, on January 30th, 2020 [3]. The transmission was monitored using a mathematical model named "The Indian Supermodel" [4, 5]. The new variant, named the "Delta variant," was discovered in India in late 2020 [6], which was the major factor leading to the second wave of infections in early April 2021 [7]. The number of active cases reached 2.5 million towards the end of April 2021, and the average number of cases reported every day touched the mark of 300,000 [8]. In early January 2022, the emergency use of vaccines like Covishield (i.e., developed by the University of Oxford) [9] and Covaxin (i.e., developed by Bharat Biotech in collaboration with the National Institute of Virology) [10] was approved by Drug Controller General of India. The Indian vaccination program began on 16 January 2021 and has crossed 1.8 billion cumulative dosages as of 14th April 2022 [11].

The double vaccine dosage system greatly decreased the number of COVID-19 cases. Still, after the advent of Omicron (a variant of SARS-CoV-2), it has been observed that many people have contracted corona even after proper vaccination. Due to the single-stranded RNA genome of SARS-CoV-2 has a high intrinsic mutation rate, which leads to immunological escape that can quickly spread in the population that has received vaccinations [12]. Omicron also displayed remarkable immunological escape power compared to the other Variants of Concern (VOC). The Omicron was first identified in November 2021 in Gauteng province, South Africa was promptly classified as a VOC on November 26, 2021 [13, 14]. This forced other nations in the world to take strict actions to stop the spread of this new deadly variant. After its identification in South Africa, this VOC rapidly disseminated to various parts of the world. According to Report 49 from the Imperial College London COVID-19 response team, the Omicron variation has a 5.4 times higher risk of reinfection than the Delta version. This suggests that the protection against reinfection provided by prior infection against Omicron reinfection may be as low as 19% [15].

The number of mutations found in the omicron variant is much higher than in other variants. The omicron variant had 32 mutations, twice the number of mutations in the Delta variant. These mutations were connected to increased transmissibility or immune evasion and had been discovered in variations including Delta and Alpha [16]. The spike (S) protein (site for antibody binding) has passed through these alterations, which produced the Omicron strain in its high infectivity and transmissibility traits. There is still much to be clarified, despite numerous investigations having sought to comprehend this new obstacle in the COVID-19 strains [17]. A new argument has emerged in favor of natural vaccinations in response to the distinctive identification of the Omicron variation. Omicron is comparable to live attenuated vaccines in several ways, which leads a number of specialists to assume that it could function as a natural vaccination. The high rate of antibody production in those who had recovered from Omicron was also emphasized as supporting evidence for the hypothesis put forth by some researchers that Omicron functions as a natural vaccine. There have also been some controversies since, like the earlier variations, it has serious health consequences and a high infection rate. However, this idea was opposed by some experts [18]. According to recent research, Omicron is remarkably resistant to the neutralizing effects of vaccinations, convalescent serum, and the majority of antibody treatments. Additionally, while most antiviral medications under research are effective against Omicron, only a small number of neutralizing antibodies are potent against it [19].

It is observed that a lot of work has been done in analyzing Chest X-Ray images using CNNs to classify the Coronavirus disease. These analyses are majorly focused on classifying

the COVID-19 outbreak. Hence, there is a need to develop a model which trains on historic COVID-19 data and accurately predicts the future of the disease compared to existing methods. The main aim of this proposed study is to provide a future prediction of COVID-19-confirmed cases using time-series data. This model will be able to accurately predict COVID-19 cases beforehand, which could help healthcare departments to take the required measures to arrange medical supplies. This could prevent unwanted loss of lives that are encountered due to a lack of infrastructure. The proposed model is based on RNN architecture using GRU and LSTM units. Furthermore, the power of residual learning is also exploited, which ultimately delivers better results when compared to other SOTA methods.

This paper is further classified into different parts. Section 2 discusses the different researcher's views and analyses of Omicron using different techniques. Section 3 discusses the data discerption and methodology used for analysis. The experiment results analysis over Omicron after vaccination is discussed in section 4. Finally, this paper is concluded in section 5 with its future aspects.

## 2. Literature review

Medicine and epidemiology have extensively used Machine Learning (ML) and DL techniques in the past few years. The previous research work done by researchers is extensively discussed below.

In [20], the authors gave a comparison of soft computing and ML models to anticipate the COVID-19 outbreak as an alternative to the SIR (Susceptible-Infected-Recovered) and SEIR (Susceptible-Exposed-Infectious-Removed) models. They showed Adaptive Network-Based Fuzzy Inference System (ANFIS), and Multi-Layered Perceptron (MLP) gave the best results. Further, they suggested ML is a potential tool to predict this outbreak as it is highly complex in nature. In [21], the authors used a DCNN model based on class decomposition known as "Decompose, Transfer, and Compose (DeTraC)" for the classification of COVID-19 in chest X-ray images. This model was applied to enhance the functionality of previously trained models. It added a class decomposition layer to the pre-trained models and considered each subset an independent class. As a result, the image dataset's classes were broken down into multiple sub-classes, which were then reassembled to provide the final predictions. This allowed for the classification of medical images under the limited availability of annotated medical images, and hence the model achieved an accuracy of 93.1%. In [22], the author used a DCNN-based model (called Inception V3) with transfer learning to detect COVID-19 automatically from chest X-Ray images. The model achieved a validation accuracy is 93%. In [23], the authors compared recent DCNN algorithms such as VGG16, VGG19, DenseNet201, InceptionResNetV2, InceptionV3, Resnet50, and MobileNetV2 for the classification of X-Ray images to detect and classify coronavirus pneumonia. Fine-tuned versions of the DCNN mentioned above models were also designed in addition to weight decay and L2-regularizers, which were used to avoid over-fitting. These models were also tested on a CT dataset for multiclass classification. Finally, it was found that InceptionResNetV2 and DenseNet201 provided better results as compared to other models, with an accuracy of 92.18% and 88.09%, respectively. In [24], the authors presented an alternative modeling framework to CNNs for identifying positive COVID-19 cases based on Capsule Networks (CapsNets), which can handle smaller datasets. CapsNets are alternative models that use 'routing by agreement' to capture spatial information. The proposed framework, termed 'COVID-CAPS,' achieved an accuracy of 95.7%, an Area Under the Curve (AUC) of 0.97, and sensitivity and specificity of 90% and 95.8%, respectively.

In [25], the authors discussed that AI could easily analyze symptoms and detect warning signs, thus helping in early detection, diagnosis of the infection, and monitoring. It can track

the spread of the virus to a certain extent at different scales, such as epidemiological, medical, and molecular scales. AI helps in projecting mortality and active cases in any region. It can also be used in drug development and diagnostic test design, helping accelerate vaccine development and making clinical trials safer. In [26], the authors proposed a procedure for deriving features for the training of Res-RNN to predict confirmed cases of COVID-19 based on meteorological factors like humidity and temperature. The proposed procedure decomposes all features and the signal to be predicted into its stationary and non-stationary modes, which were used to train separate Res-RNN. The results were summed to derive the final forecast of COVID-19 cases. A MAPE value of 4.68% was achieved, which confirms the applicability of the technique. In [27], the authors developed a nature-inspired Algorithm, a hybridized approach where an enhanced version of the Beetle Antennae Search (BAS) algorithm was used to improve the prediction model performance and to determine the parameters of the ANFIS. Antecedent and conclusion ANFIS parameters were taken into consideration. This method was then evaluated using WHO's official data on the COVID-19 outbreak.

## 3. Proposed methodology and data set

This section discusses the dataset description and proposed model formulation for analyzing the impact of Omicron after vaccination.

### 3.1. Dataset description

The data is collected from the HDX Novel Coronavirus (COVID-19) Cases dataset, available online [28]. This data has been recorded since 22 January 2020 and is categorized country-wise, which is updated daily and maintained by Johns Hopkins University Centre for Systems Science and Engineering (JHU CSSE). The data is divided into the following categories: confirmed cases, vaccinated cases, recovered cases, unconfirmed cases, and daily tweet volume (which shows the change in the number of cases with several tweets). The data were divided into training and validation, 90% and 10%, respectively. The details about the dataset are given in Table 1.

### 3.2. Proposed methodology

This work proposes an RNN technique-based model that is prominent in handling sequential time series data. While training the RNN network, we encounter a major problem called vanishing gradients as the network goes deeper. This problem arises due to the continuous multiplication of weights while Backpropagation Through Time (BPTT) with increasing requirements of learning long-term dependencies. GRU and LSTM units were introduced into the network to solve this problem, as shown in Fig 1. These units focus on the important features that must be carried forward in the network. After using the GRU and LSTM units, the model faced the problem of learning from repeated data. It is very common for a time series dataset to have repeating data. As the GRU units tend to learn from repeated data, it becomes extremely harmful for the model as the model tends to get biased for a particular data point. Hence, a residual architecture is used in the network, combining the power of Residual

**Table 1. Description of the dataset used.**

| Dataset | Number of rows | Number of columns | Training | Testing |
|---|---|---|---|---|
| HDX Covid-19 Dataset | 918 | 2 | 818 | 100 |

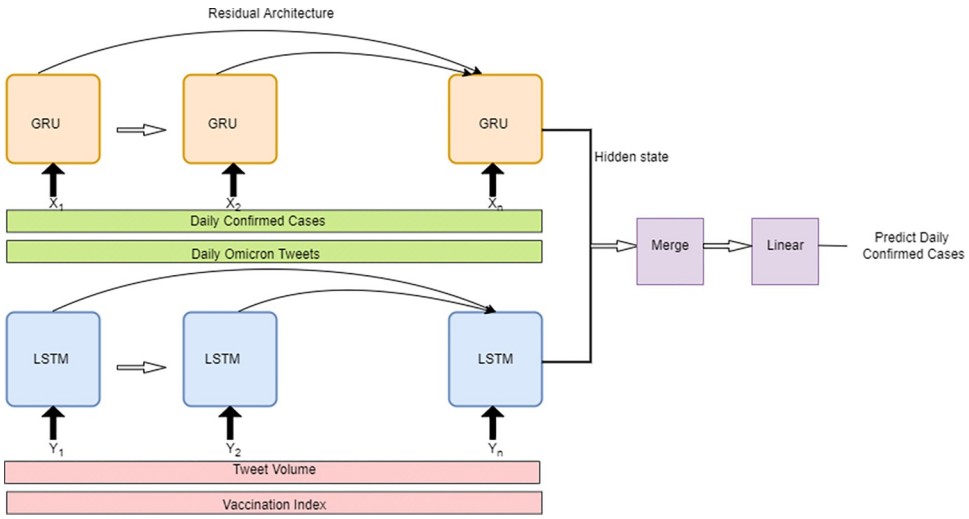

**Fig 1. Framework architecture.**

architecture aided with GRU and LSTM units. These units are used as gates to regulate which input is being transferred to the residual network.

As mentioned, BPTT updates the gradients while learning long-term dependencies in RNNs. BPTT is similar to the backpropagation used in standard neural networks and CNNs. Because of continuous multiplication, the weights converge to near zero, preventing the model from learning or updating itself. The problem of vanishing gradients is also addressed using GRU units which handle it efficiently. GRU consists of two gates, i.e., update and resets gates, respectively. The 1$^{st}$ gate determines what amount of information is required to pass into the next unit, while the 2$^{nd}$ gate decides how much information is required to erase. In this way, GRU helps keep only a certain amount of important information. The different gates in GRU are described in Eq (1):

$$p = \sigma(W_p x_t + U_i a_{t-1})$$

$$q = \sigma(W_q x_t + U_q a_{t-1})$$

$$r = \tanh(W_r x_t + U_r(a_{t-1} \odot q))$$

$$a_t = (1 - p) \odot r + p \odot a_{t-1} \tag{1}$$

Here, $p$ and $q$ refer to reset and update gates, respectively. $W$ is the weighted parameter, $a$ is the hidden state, and $\odot$ refers to element-wise multiplication.

Next, Residual learning is used to reduce redundancy in learning. Residual Networks have recently achieved prominent results in developing CNN models and hence are introduced in RNNs too. Hence, residual links have been added to the GRU units. These residual links extract the unique information from the data and forward it to the network. The shortcut connections or the residual links do not add extra parameters and hence do not increase the complexity of the model. Finally, the output from the GRU and LSTM units is guided to a merged layer that assembles the output from the corresponding GRU units and the output from the tweet volume. The result is then passed through a linear layer where the assembled output provided by the merged layer is normalized. Residual Recurrent Network (RRN) is used in the

model, as shown in Fig 1. The basic architecture of RRN is very similar to a simple RNN. Let each hidden state be represented by $a = \{a_1,\ldots,a_T\}$, $T$ being steps of hidden states and sequential input be represented by $x = \{x_1,\ldots,x_T\}$. Identity connections are made from $a_{t-1}$ unit to $a_t$ unit and recurrent transformation also takes place, which takes $a_t$ and $x_t$ as input, then recurrent transformation takes place in the form given in Eq (2).

$$(a_t = (RRN(x_t, a_{t-1}))$$

$$a_t = h(g(a_{t-1}) + F(a_{t-1}, x_t; W)) \tag{2}$$

Here, $F$ is the residual function, and $W$ is the weighted parameters. $g(a_{t-1})$ is the identity function. h is the activation function which is traditionally set as tanh. As higher recurrent depth leads to better performance, the depth is taken as K. The state-wise output is shown in Eq (3):

$$y_1^t = \sigma(x_t W_1 + a_{t-1} U_1 + b_1)$$

$$y_2^t = \sigma(x_t W_2 + y_1^t U_2 + b_2)$$

$$\ldots$$

$$y_K^t = \sigma(x_t W_K + y_{K-1}^t U_K + b_k)$$

$$F(a_{t-1}, x_t) = y^t \tag{3}$$

Also, $x_t$ is taken at every layer.

LSTM is based on the basic structure of RNN, which avoids gradient vanishing and is a powerful tool for remembering long-range dependencies. At each time step $t$, inputs ($x_t, c_{t-1}, a_{t-1}$) are sent to three gates, namely, the input gate, output gate, and forget gate ($c_{t-1}$ denotes previous memory state) which generates three signals, $i_t, o_t, f_t$ respectively. The mathematical formulation is shown in Eq (4)

$$f_t = \sigma(x_t * U_f + a_{t-1} * W_f)$$

$$i_t = \sigma(x_t * U_i + a_{t-1} * W_i)$$

$$o_t = \sigma(x_t * U_0 + a_{t-1} * W_o)$$

$$\tilde{c}_t = \tanh(x_t * U_c + a_{t-1} * W_c)$$

$$c_t = f_t * c_{t-1} + i_t * \tilde{c}_t \tag{4}$$

## 3.3. Metrics used

The performance of the proposed model is analyzed based on different performance matrices such as True Positive (TP), False Positive (FP), True Negative (TN), and False Negative (FN). The accuracy, precision, recall, Negative Predictive Value, and sensitivity of the model are calculated using Eq (5) to Eq (9).

$$Precision = \frac{TP}{(TP + FP)} \tag{5}$$

$$Recall = \frac{TP}{(TP + FN)} \tag{6}$$

$$Negetive\ Predictive\ Value = 2 * \frac{Precision * Recall}{Precision + Recall} \tag{7}$$

$$Accuracy = \frac{TP + TN}{(TP + TN + FP + FN)} \tag{8}$$

$$Specificity = \frac{TN}{(TN + TP)} \tag{9}$$

Mean Squared Error (MSE) gives a better idea when dealing with time-series or regression tasks. MSE is the mean of the square of the error between the predicted value and the ground truth as shown in Eq (10).

$$MSE = \frac{1}{N} \sum_{i=1}^{N} (y_i - \hat{y}_i)^2 \tag{10}$$

## 4. Experimental results and analysis

The results of the proposed RNNCON-Res and SOTA methods (i.e., RNN and Res-RNN) are presented and compared in this section. Res-RNN provided better results when compared to simple RNNs as the residual links in the network help in focusing important features in the input as a shortcut connection carries the information over the network. When the proposed model is compared with the Res-RNN model, it outperforms in terms of accuracy due to the involvement of the GRU unit in the network, as it solves the problem of vanishing gradients while remembering the long-term dependencies. The covid-19 dataset is combined with tweet volume as an additional feature used to reach data augmentation. The proposed model provides better results when compared with SOTA methods, as shown in Fig 2.

**The Autoregressive Model (AR)** model is very popular for forecasting time series data using a linear combination of past values of the variable [29]. An order-p autoregressive model can be mathematically described as shown in Eq (11):

$$x_t = \alpha_p x_{t-p} + \varepsilon_t + c \tag{11}$$

The prediction for a given time-stamp *t* is a weighted sum of past data points in a given period of size '*w*' multiplied by a parameter $a_p$. White noise is introduced, which is a small random noise represented by $\varepsilon_{t+h}$. The deviation between the true and linear values can be explained using this. *c* is intercept s. **The GAR is a model which** is trained with one set of $\alpha_p$ and c for all the sources when the signal received from all the different sources shows the same patterns and data required to train the system is limited. **Vector Autoregression (VAR)** is a forecasting algorithm used in multivariate time series models, as shown in Eq (12):

$$\bar{x}_{t+h} = \sum_{p=0}^{w-i} A_p\ x_{t-p} + \varepsilon_{t+h} + c \tag{12}$$

Where signal-wise $\alpha_p$ in AR is replaced with a matrix $A_p$ to capture the correlation information. Fig 3 shows the comparison of RNNCON-Res with ResRNN. In this, ResRNN can predict the rise and fall of the prediction line of the graph but fails to give accurate numbers of covid-

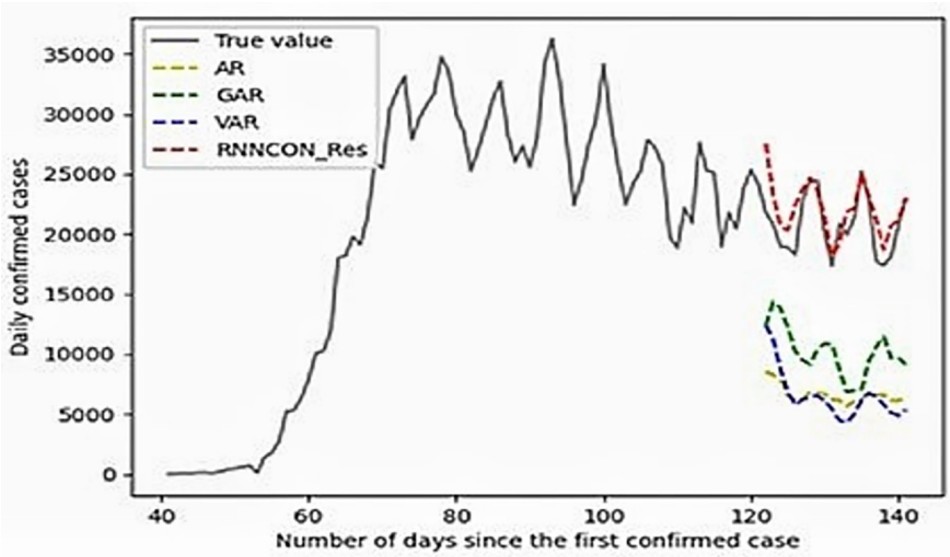

**Fig 2. Comparison of the number of covid case predictions using RNNCON-Res and other models.**

19 cases. However, RNNCON-Res can predict accurately. Finally, it is concluded that the results of the proposed model are better than other existing models.

The proposed model is also compared with ARIMA and FBProphet time series models, as shown in Fig 4. It can be observed that ARIMA performs better in predicting the covid-19 cases compared to FBProphet. However, these models do not perform better than the proposed RNNCON-Res model.

The 91% accuracy has been achieved using the RNNCON-Res model, as discussed in Fig 5. As RNNCON-Res gave promising results, Exploratory Data Analysis d(EDA) was performed to plot the graph of the number of probable cases based on ground truth. Fig 6(A) shows the

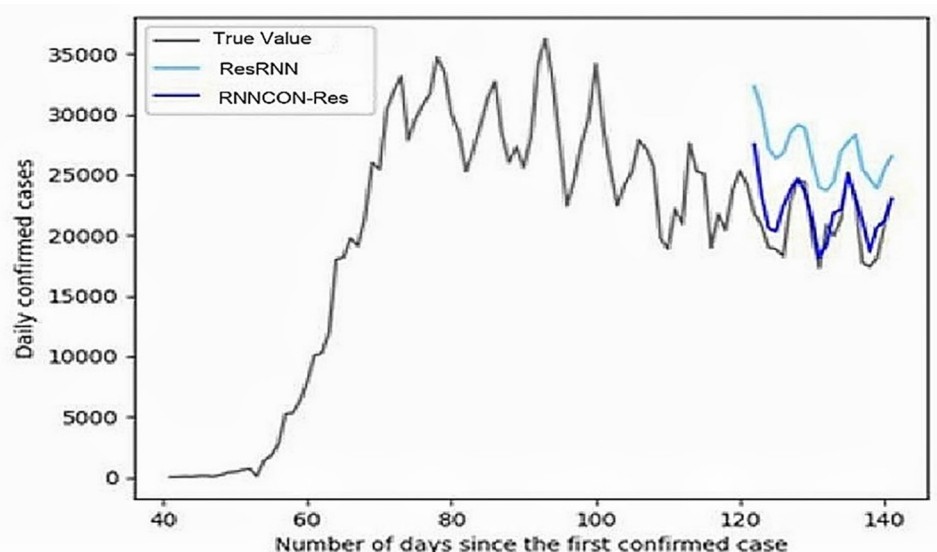

**Fig 3. Comparison of the number of covid case predictions using RNNCON-Res and ResRNN with true value.**

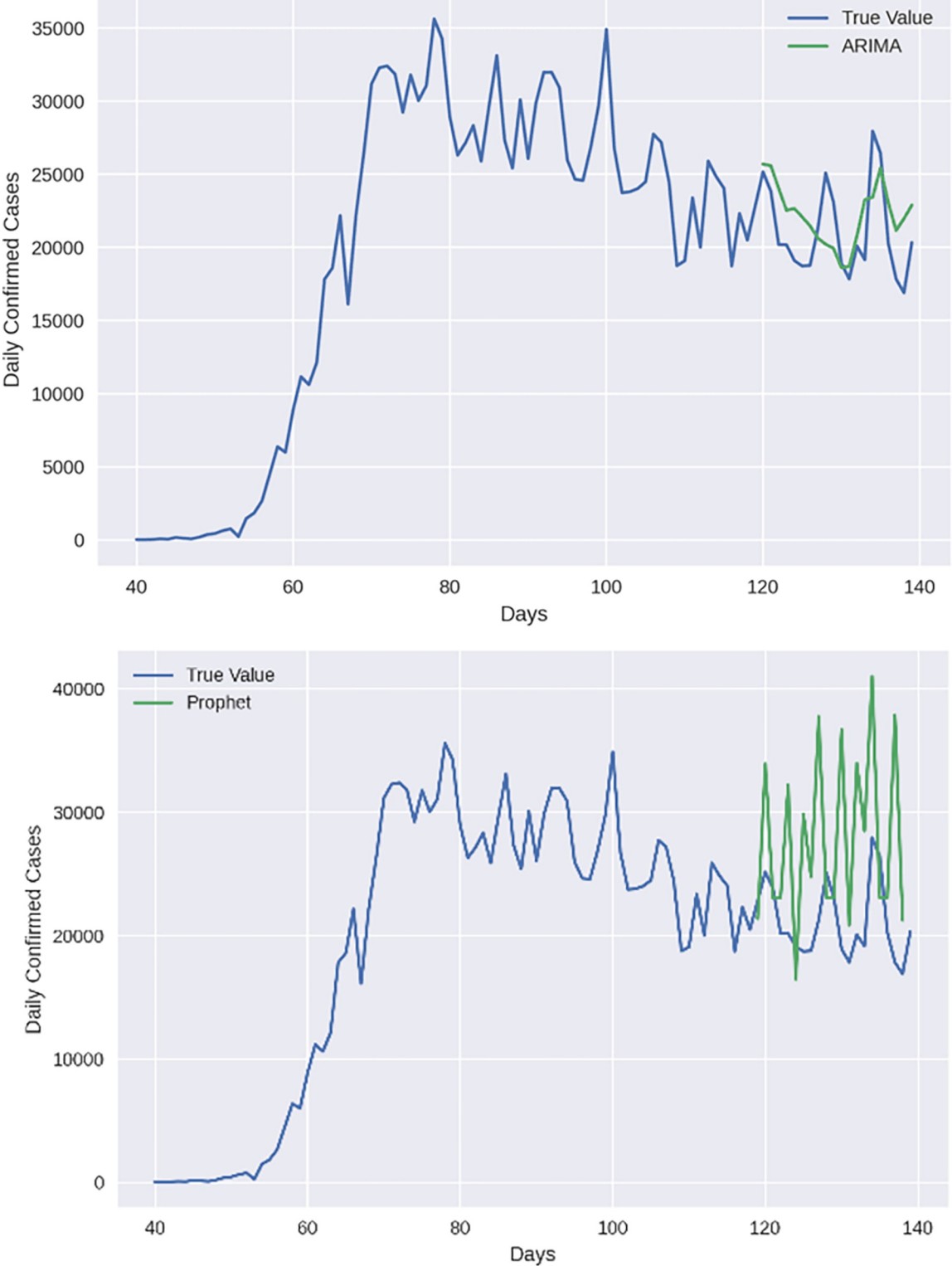

**Fig 4.** Comparison of the number of covid-19 cases prediction using (a) ARIMA and (b) FBProphet with True Value.

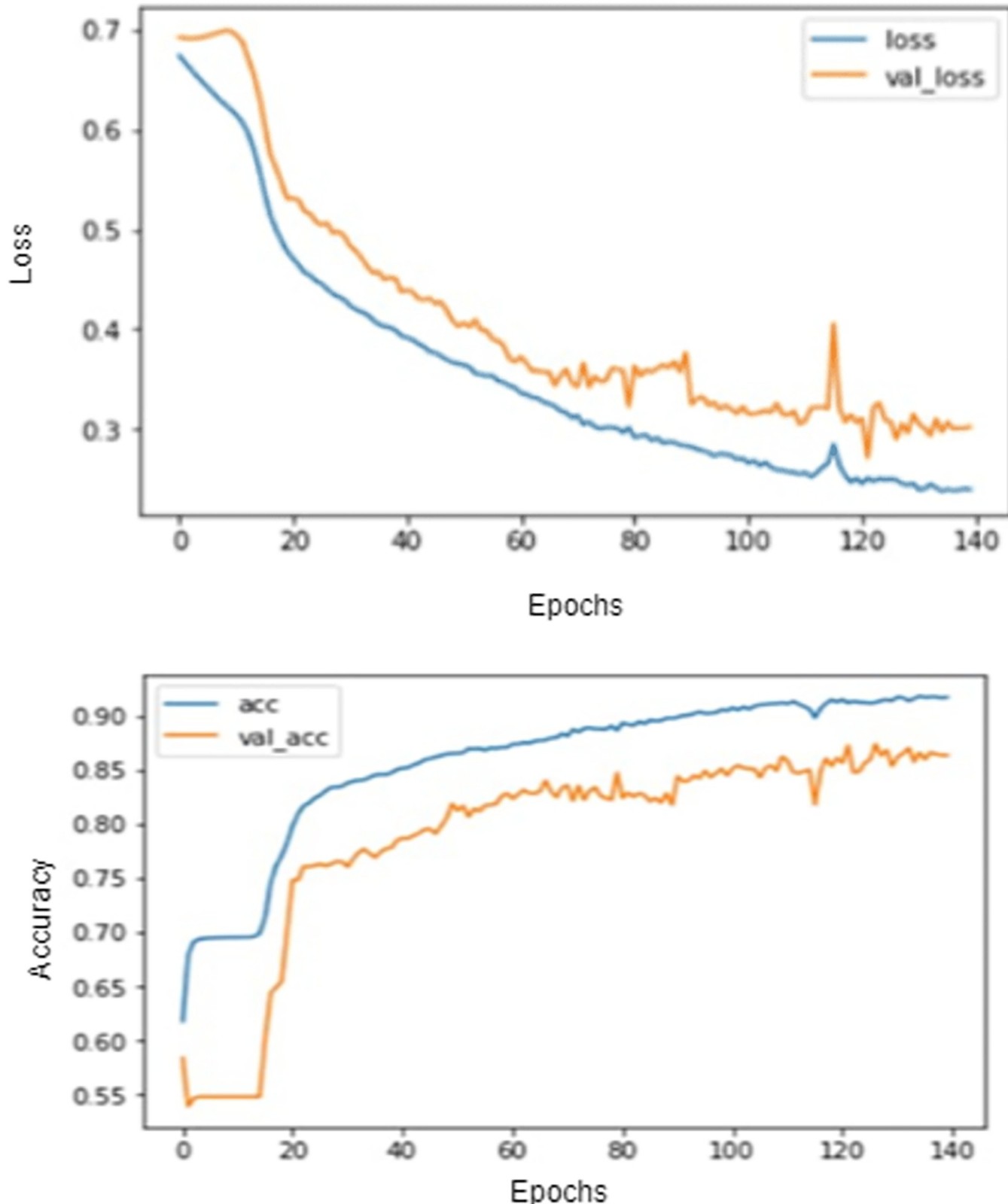

**Fig 5. Graphs for Training and validation for RNNCON-Res.** (a) loss (b) accuracy.

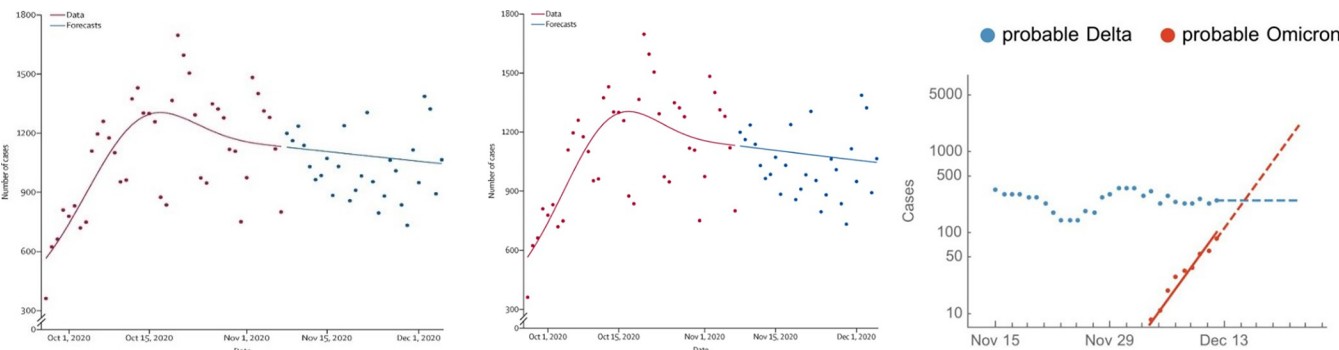

**Fig 6.** Predictions based on RNNCON-Res (a) Shows total probable COVID-19 cases based on ground truth (b) Shows probable Omicron and Delta variant cases.

total covid-19 cases with the ground truth values. Based on the present scenario, the model can forecast the total number of cases up to 20 days.

The probable cases of each Omicron and Delta variant were also estimated using the model shown in Fig 6(B) and the total number of recovered cases over the total number of covid-19 cases as shown in Fig 7.

The exact details for the performance of the proposed model and the results generated for different classes are better described using a confusion matrix, as shown in Fig 8. It shows individual values of different metrics given in Eq (5) to Eq (9). The value achieved for Precision, Negative Predictive Value, specificity, sensitivity, and accuracy are 0.904, 0.926, 0.914, 0.917, and 91.6% respectively.

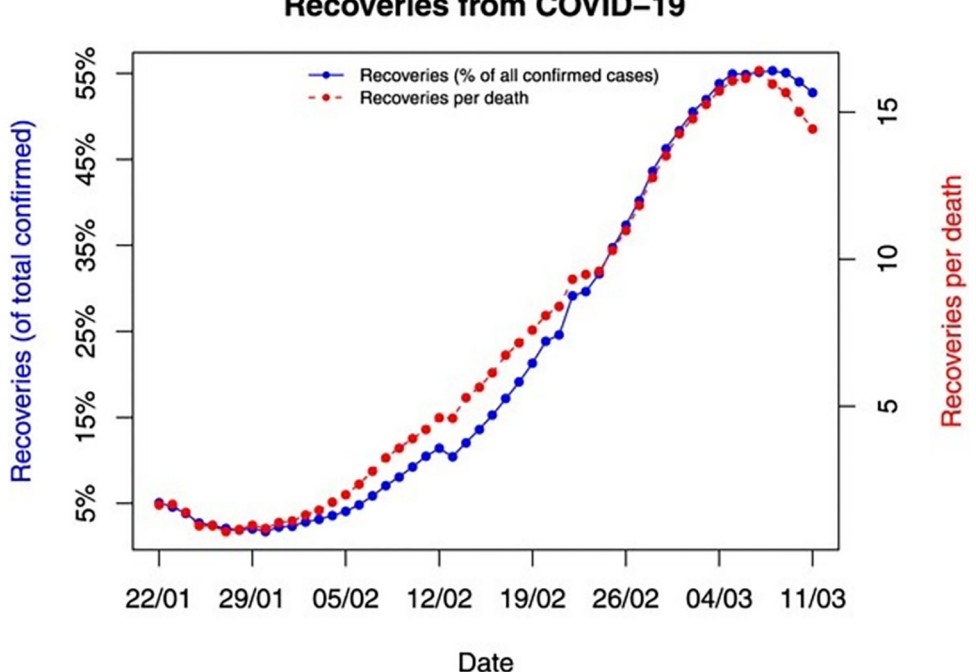

**Fig 7. Total number of confirmed, death, and recovered cases.**

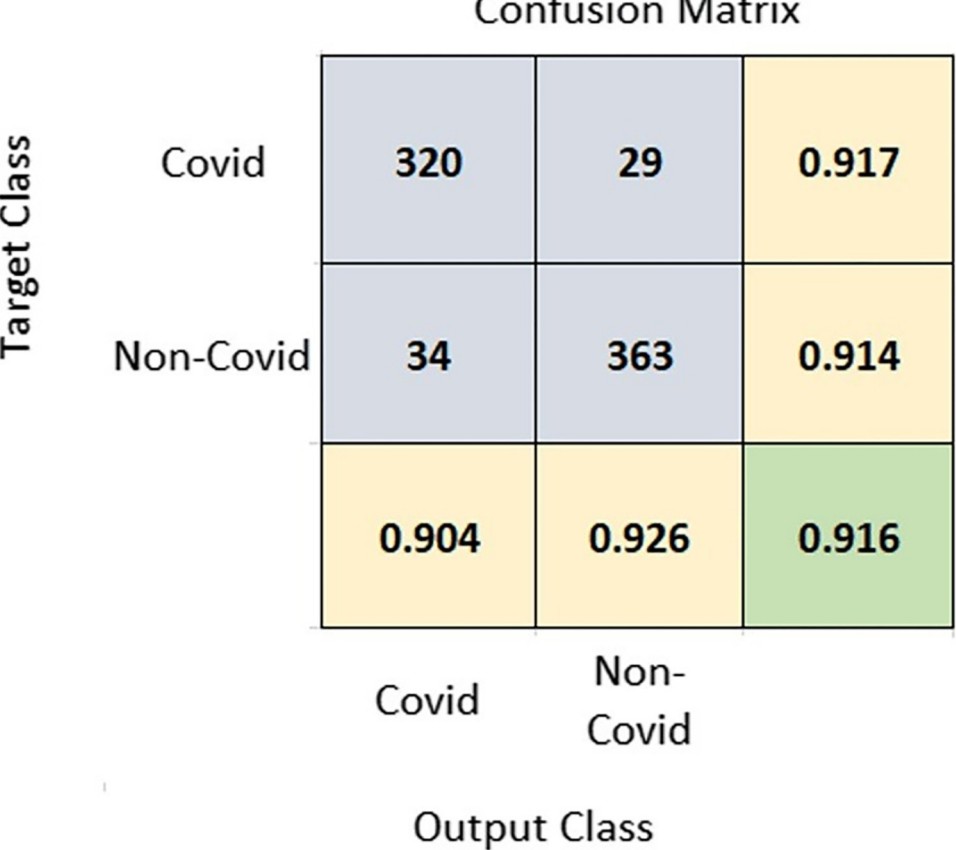

**Fig 8. Confusion matrix for result analysis of RNNCON-Res.**

## 5. Discussion of performance

The results received from RNNCON-Res are compared with other methods, such as Recurrent Neural Model [30] and Stacked-LSTM [31], as discussed in Table 2. Stacked-LSTM is advanced by using LSTM units instead of simple recurrent units used in the Recurrent Neural Model for better results. 90% accuracy was achieved using Stacked-LSTM. In the proposed model, LSTM and GRU units are used along with a residual network, which provides better results than Stacked-LSTM and achieved the benchmark of 91% accuracy.

The MSE for RNNCON-Res is calculated to be 4067567.11. The model is also compared with other models like ARIMA [32], BSTS [32], and NAR Neural Model [33]. The ARIMA and BSTS model delivered RMSE of 4391 and 3874 on the HDX dataset, and the NAR Neural Model delivered RMSE of 47366 on the entire data. RNNCON-Res performed the best among these models reaching RMSE of 2016.

**Table 2. Comparison of accuracy delivered by RNNCON-Res with other models.**

| S. No | Model | Dataset used | Accuracy (in %) |
|---|---|---|---|
| 1. | RNNCON-Res (Proposed) | HDX | **91** |
| 2. | Recurrent Neural Model | UCI ML Repository | 88 |
| 3. | Stacked-LSTM | HDX | 90 |

## 6. Conclusion and future scope

Predicting disease propagation is an essential part of disease management and control. Some successfully applied models include ANFIS, MLP, DeTraC, VGG16, VGG19, etc. But, in many cases, predictive neural networks may struggle due to small datasets. This paper predicted the outbreak of COVID-19 in the US using historic daily confirmed cases and Twitter data. The prediction capabilities of various powerful models such as the AR, GAR, Recurrent Neural Model, Stacked-LSTM, and Res-RNN were taken into consideration while predicting the outbreak of COVID-19 in the US using historic daily confirmed cases and Twitter data, which did not give very accurate results. The proposed model uses GRU and LSTM units with Residual links to tackle the over-fitting problem and to remember long-term dependencies. Data augmentation was facilized by introducing a merged layer before the linear layer to use tweet volume as an additional feature (as neural networks are known to require huge data). The RNNCON-Res model demonstrated dominating capability in country-level prediction 20 days ahead. Also, RNNCON-Res can be used to solve any time series forecasting problem. In the future, this model can be used in reinforcement learning with sequential data, making it a self-learning machine to better predict the spread of COVID.

## Author Contributions

**Conceptualization:** Meenu Gupta.

**Formal analysis:** Rakesh Kumar, Aman Agarwal, Anustup Mukherjee.

**Methodology:** Meenu Gupta.

**Software:** Aman Agarwal.

**Supervision:** Rakesh Kumar.

**Validation:** Sardar M. N. Islam.

**Visualization:** Sardar M. N. Islam.

**Writing – original draft:** Rakesh Kumar, Meenu Gupta.

**Writing – review & editing:** Sardar M. N. Islam.

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
