## [Decision Letter · Decision Letter 0]

4 Nov 2022

PONE-D-22-26738Epidemic efficacy of Covid-19 vaccination against Omicron: An Innovative Approach using Enhanced Residual Recurrent Neural NetworkPLOS ONE

Dear Dr. Gupta,

Thank you for submitting your manuscript to PLOS ONE. After careful consideration, we feel that it has merit but does not fully meet PLOS ONE’s publication criteria as it currently stands. Therefore, we invite you to submit a revised version of the manuscript that addresses the points raised during the review process.

We look forward to receiving your revised manuscript.

Kind regards,

Tarik A. Rashid, PhD

Academic Editor

PLOS ONE

**Journal Requirements:**

2. Please note that PLOS ONE has specific guidelines on code sharing for submissions in which author-generated code underpins the findings in the manuscript. In these cases, all author-generated code must be made available without restrictions upon publication of the work. Please review our guidelines at https://journals.plos.org/plosone/s/materials-and-software-sharing#loc-sharing-code and ensure that your code is shared in a way that follows best practice and facilitates reproducibility and reuse. New software must comply with the Open Source Definition.

4. Please upload a new copy of Figures 2 and 3 as the detail is not clear. Please follow the link for more information:

https://blogs.plos.org/plos/2019/06/looking-good-tips-for-creating-your-plos-figures-graphics/

https://blogs.plos.org/plos/2019/06/looking-good-tips-for-creating-your-plos-figures-graphics/

Reviewers' comments:

Reviewer's Responses to Questions

**Comments to the Author**

1. Is the manuscript technically sound, and do the data support the conclusions?

Reviewer #1: Yes

Reviewer #2: Yes

2. Has the statistical analysis been performed appropriately and rigorously? 

Reviewer #1: Yes

Reviewer #2: Yes

3. Have the authors made all data underlying the findings in their manuscript fully available?

Reviewer #1: Yes

Reviewer #2: Yes

4. Is the manuscript presented in an intelligible fashion and written in standard English?

Reviewer #1: Yes

Reviewer #2: No

5. Review Comments to the Author

Reviewer #1: The manuscript entitled “Epidemic efficacy of Covid-19 vaccination against Omicron: An Innovative Approach using Enhanced Residual Recurrent Neural Network” presents an enhanced version of Residual Neural Network using Gated Recurrent Unit (GRU) and LSTM units to train a reliable model for predicting the spread of Coronavirus.

Considering the great effort done by the authors, there are some issues as follows:

1) The structure of the of introduction can be improved. It is suggested to remove subsections of section 1 for an integrated text. The aim of the paper has been stated before section 1.1, and again, in section 1.2 and 1.3.

2) Please check the journal guidelines whether the first letter of the authors’ names should be written in the citations within text, or not. A modification maybe required.

3) The Comparison and reliability related results cab be extended. More accurate and competing methods should be further utilized in the result comparison. The prediction of basic ResRNN is more accurate than AR, GR, VAR. So, using these methods for comparison is not wise.

4) Identify the names of the axis in Fig.4.

5) The models can also be compared using the Mean Squared Errors.

6) Its better to present the prediction results (such as Fig. 6) considering the competitive results of the other methods.

7) It is suggested to add the pseudo code of the proposed method.

Considering the mentioned comments, the paper is required a major revision.

Reviewer #2: Thanks for the submission,

The language needs proofreading and please provide a proof if you already done this.

In 1.1 how would you proof that other previous works are limited? Could you name any research?

In 1.3 what do you mean by contribution? change the title or give more clear details please.

Have you reviewed any previous literature regarding omicron against Covid-19 vaccination? I could not find any. Please name some or highlight them for me.

There is not any table containing the dataset and your description about the columns and rows you used in your research.

The discussion part is too short.

May we have any future work regarding omicron? what would you do in the future about it?

Kind Regards

6. PLOS authors have the option to publish the peer review history of their article (what does this mean?). If published, this will include your full peer review and any attached files.

Reviewer #1: **Yes: **Salar Farahmand-Tabar

Reviewer #2: **Yes: **Askandar H. Amin

---

## [Author Response · Author response to Decision Letter 0]

12 Dec 2022

Dear reviewers,

We are thankful to you for reading our manuscript and providing valuable suggestions to improve it. The response with respect to queries asked by reviewers is mentioned below and highlighed in the manuscript.

Reviewer #1

Question 1: The structure of the of introduction can be improved. It is suggested to remove subsections of section 1 for an integrated text. The aim of the paper has been stated before section 1.1, and again, in section 1.2 and 1.3.

Response: The structure of Introduction has been changed and the sub-sectioning has been removed for an integrated text.

Question 2: Please check the journal guidelines whether the first letter of the authors’ names should be written in the citations within text, or not. A modification maybe required.

Response: The author’s names have been updated according to requirements

Question 3: The Comparison and reliability related results cab be extended. More accurate and competing methods should be further utilized in the result comparison. The prediction of basic ResRNN is more accurate than AR, GR, VAR. So, using these methods for comparison is not wise.

Response: The data has been trained on two more models namely, ARIMA and FB Prophet which are quite competing methods in time-series prediction and its result graphs have been added (refer Fig 4)

Question 4: Identify the names of the axis in Fig.4.

Response: The names of the axis in Fig 4(Now Fig 5) have been added

Question 5: The models can also be compared using the Mean Squared Errors.

Response: The Mean Squared Error (MSE) for the proposed model and some other models has been added.

Question 6: Its better to present the prediction results (such as Fig. 6) considering the competitive results of the other methods.

Response: The Fig 6 (Now Fig 7) is an Exploratory Data Analysis and comparision of this with other model is not applicable.

Question 7: It is suggested to add the pseudo code of the proposed method.

Response: We can provide the code as per requirement after acceptance. 

Reviewer #2

Question 1: The language needs proofreading and please provide a proof if you already done this.

Response: The entire manuscript has been proofread carefully and the changes have been highlighted.

Question 2: In 1.1 how would you proof that other previous works are limited? Could you name any research?

Response: Most of the work regarding to COVID-19 has been done is classifying whether a person has COVID-19 or not using images. There are few articles on predicting COVID-19. Moreover, the architecture designed by us using residual architecture with LSTMS and GRU units has not been implemented yet.

Question 3: In 1.3 what do you mean by contribution? change the title or give more clear details please.

Response: The sub-sections in introduction has been removed as guided by reviewer 1.

Question 4: Have you reviewed any previous literature regarding omicron against Covid-19 vaccination? I could not find any. Please name some or highlight them for me.

Response: Yes, we have reviewed article for omicron against covid-19 vaccination and it has been highlighted in references section

Question 5: There is not any table containing the dataset and your description about the columns and rows you used in your research.

Response: The table containing the dataset description has been added (refer table 1). 

Question 6: The discussion part is too short.

Response: More information about model performance based on MSE and RMSE is added in discussion

Question 7: May we have any future work regarding omicron? what would you do in the future about it?

Response: Future work is mentioned in the conclusion and highlighted for your reference

---

## [Editor Report · Decision Letter 1]

21 Dec 2022

Epidemic efficacy of Covid-19 vaccination against Omicron: An Innovative Approach using Enhanced Residual Recurrent Neural Network

PONE-D-22-26738R1

Dear Dr. Gupta,

We’re pleased to inform you that your manuscript has been judged scientifically suitable for publication and will be formally accepted for publication once it meets all outstanding technical requirements.

Kind regards,

Tarik A. Rashid, PhD

Academic Editor

PLOS ONE

Additional Editor Comments (optional):

Make sure to cite all the equations that are not yours.

The quality of the figures need to be improved.
---

## [Editor Report · Acceptance letter]

1 Feb 2023

PONE-D-22-26738R1 

Epidemic efficacy of Covid-19 vaccination against Omicron: An Innovative Approach using Enhanced Residual Recurrent Neural Network 

Dear Dr. Gupta:

I'm pleased to inform you that your manuscript has been deemed suitable for publication in PLOS ONE. Congratulations! Your manuscript is now with our production department. 

Kind regards, 

on behalf of

Dr. Tarik A. Rashid 

Academic Editor

PLOS ONE